# Single and Combined Salinity and Heat Stresses Impact Yield and Dead Pericarp Priming Activity

**DOI:** 10.3390/plants10081627

**Published:** 2021-08-08

**Authors:** Bupur Swetha, Jeevan R. Singiri, Nurit Novoplansky, Rohith Grandhi, Jansirani Srinivasan, Janardan Khadka, Ivan Galis, Gideon Grafi

**Affiliations:** 1French Associates Institute for Agriculture and Biotechnology of Drylands, Jacob Blaustein Institutes for Desert Research, Ben-Gurion University of the Negev, Midreshet Ben Gurion 84990, Israel; bupurswethanagaraju1234@gmail.com (B.S.); jeevannaveen01@gmail.com (J.R.S.); nuritnov@bgu.ac.il (N.N.); rohithvyshnavi017@gmail.com (R.G.); jansirani085@gmail.com (J.S.); janardankhadka@gmail.com (J.K.); 2Institute of Plant Science and Resources, Okayama University, Kurashiki 710-0046, Japan; igalis@rib.okayama-u.ac.jp

**Keywords:** dead pericarps, salinity, short episodes of high temperature, combined stresses, priming, reproductive phase, seed abortion, phytohormones, *Brassica juncea*

## Abstract

In the face of climate change and the predicted increase in the frequency and severity of abiotic stresses (e.g., hot spell, salinity), we sought to investigate the effect of salinity (S), short episodes of high temperature (HS) and combination of salinity and high temperature (SHS), at the reproductive phase, on yield with a special focus on the properties of dead pericarps of *Brassica juncea*. Three interval exposures to HS resulted in massive seed abortion, and seeds from salt-treated plants germinated poorly. Germination rate and final germination of *B. juncea* seeds were slightly reduced in the presence of salt and SHS pericarp extracts. All pericarp extracts completely inhibited seed germination of tomato and *Arabidopsis,* but removal of pericarp extracts almost fully restored seed germination. Heat and salinity profoundly affected the accumulation of phytohormones in dead pericarps. Combined stresses highly reduced IAA and ABA levels compared with salt, and enhanced the accumulation of GA1, but abolished the positive effect of salt on the accumulation of GA4, JA and SA. Interestingly, pericarp extracts displayed priming activity and significantly affected seedling performance in a manner dependent on the species and on the origin of the pericarp. While control pericarps improved and reduced the seedlings’ performance of autologous and heterologous species, respectively, pericarps from salt-treated plants were harmless or improved heterologous seedling performance. Thus, the strategy employed by the germinating seed for securing resources is set up, at least partly, by the mother plant in conjunction with the maternal environment whose components are stored in the dead maternal organs enclosing the embryo.

## 1. Introduction

Abiotic stresses, which are likely to increase in severity and frequency as a result of global climate change pose an acute threat on food security worldwide. Accordingly, abiotic stresses such as drought, salinity, and temperature extremes have a devastating effect on the yield of major crops, which endangers food security worldwide [1,2,3,4]. Climate change might lead to soil salinization and impacts agricultural areas in arid, semi-arid and coastal regions of the world as well as in European countries, which limits the potential use of these soils in agriculture [5,6]. Climate change has a notable impact on the average annual temperature, but more importantly it can lead to extreme climate events, including heat waves and hot spell [7,8]. These heat waves are the most critical factors affecting crop yield, especially when observed together with other stresses and during the reproduction stage [9,10]. Indeed, the maternal environment is reported to be a major factor affecting yield and progeny seed properties [11,12,13].

Since the realization that plants respond uniquely to a combination of stresses [14], which is not merely the summation responses to each stress, multiple reports have been published addressing plant response to single and combined stresses. Accordingly, these reports clearly approved the idea that a combination of stresses often yields a specific response, which is not seen under either single stress condition [15,16,17,18]. Yet, a combination of stresses not only elicits a specific response but may either mitigate [19] or enhance [20,21] the stress effect.

While most studies addressing plant response to single or combined stresses focused on crop yield or on vegetative growth performance, not many studies have addressed the effect on the maternally-derived dead components of the dispersal unit (e.g., seed coat, pericarps), designated dead organs enclosing embryos (DOEEs). Recent work demonstrated that the maternal environment not only affects the embryo properties but also the properties of DOEEs [18,22,23]. Besides providing a shield for embryo protection and a means for dispersal, DOEEs also function as a long-term storage for multiple substances (proteins, metabolites) that can persist in active form for decades and affect seed germination and fate [24,25,26]. The expected changes in the DOEEs properties as a result of the maternal environment [18,22,23] can have an impact on seed viability and persistence, germination and seedling establishment and consequently on plant population dynamics and diversity [27,28,29,30].

Phytohormones represent a group of structurally unrelated substances that play a key regulatory role in plant growth and development and response to biotic and abiotic stresses [31]. DOEEs store and accumulate phytohormones whose levels are affected by the maternal environment. Pericarps of *Anastatica hierochuntica* and floral bracts (lemma and palea) of *Avena sterilis* possess multiple phytohormones including abscisic acid (ABA), auxin and particularly high levels of salicylic acid (SA), whose levels were changed under salt and drought conditions [22,23]. Likewise, phytohormone analysis of the dead glumes of *Triticum turgidun* ssp. *dicoccoides* revealed the presence of ABA, indole acetic acid (IAA) and high levels of SA and jasmonic acid (JA) [22] (Raviv et al., 2018). These stored phytohormones might play a role in seed dormancy and germination as well as in seed priming and post germination growth [31,32,33].

Abiotic stresses such as low and high temperature, salinity and drought elicit a common response in land plants and negatively impact growth and development; in crop plants they cause significant yield losses worldwide [3,34,35,36,37,38]. Multiple studies related to the effect of heat shock on plant performance were performed under long-term exposure (often >24 h) to high temperatures (37–45 °C), though in recent years the effect of short episodes of high temperature (heat waves/hot spells) are receiving more attention [17,38]. Here, we sought to examine the effect of salinity and short-term exposure to high temperature during the reproductive phase on progeny seed production and particularly the DOEE properties of *Brassica juncea* (L.) Czern & Coss. (Brassicaceae) that together with other *Brassica* species represent an important source of vegetable oil worldwide [39]. Similar to other *Brassica* and leguminous crop plants, *B. juncea* dry fruits are indehiscent, that is the fruit remains intact and does not split open at maturity. We investigated the effect of single and combined salt and heat on seed progeny production and fate with a special focus on the properties of dead pericarps. Our data demonstrated the elaborated function of *B. juncea* dead pericarps as a storage entity for beneficial substances whose levels, composition and function are significantly altered in response to single and combined stresses.

## 2. Materials and Methods

### 2.1. Plant Growth Conditions and Exposure to Stress

*Brassica juncea* (Indian mustard) seeds purchased from the local market were sown in standard gardening soil composed of peat and perlite (2:1 ratio) in small pots. Mustard seedlings were transplanted (18 days after sowing) into 1L pots having Hamra red sandy soil [40] supplemented with 4 g/L slow-release fertilizer (Green Multigan 20% N, 11% P_2_O_5_, 16% K_2_O and trace elements). Briefly, plants were irrigated every two days with tap water for one month until the beginning of bolting, at which time half of the plants were exposed to a salt stress of 50 mM NaCl for one month, after which NaCl concentration was gradually increased in a 5-day manner to 75, 100, 150 up to 200 mM. After reaching the highest salt concentration, half of the water and salt-irrigated plants were subjected to 3 intervals of heat shock treatment (37 °C, 3 h each) in a course of 5 days to obtain the moderate effect of a heatwave. According to the World Meteorological Organization, a heatwave is defined as 5 or more consecutive days of prolonged heat in which the daily maximum temperature is higher than the average maximum temperature by 5 °C (9 °F) or more (http://www.grida.no/climate/ipcc_tar/wg2/061.htm#1434; accessed on 30 October 2007). Notably, at the time of salt application, the plants were irrigated with distilled water (DW) or DW+salt. Irrigation with 200 mM NaCl was continued for another week and then all pots were irrigated with tap water until fruit matured and dried out 6 weeks later. Mustard pods were harvested and measured for fruit weight and length, seed weight, seed abortion and germination capacity as well as pericarp properties.

The response of the plants to short episodes of heat shock was confirmed by immunoblotting. Accordingly, total proteins were extracted from the leaves of the control and stress-treated plants by the NETN (20mM Tris-HCl, pH 8.0, 100 mM NaCl, 1 mM EDTA, 0.5% Nonidet P-40) buffer supplemented with a protease inhibitor cocktail (Sigma, St. Louis, MO, USA). Protein concentration was determined by the Bradford reagent (BioRad, Hercules, CA, USA) and 15 μg of total proteins resolved by SDS/PAGE and immunoblotted with rabbit polyclonal antibodies to HSP70 (AS08 371, Agrisera AB, Vannas, Sweden) and to HSP17.6 (AS07 254, Agrisera AB, Vannas, Sweden). Immuno-detection was performed using the secondary antibody of goat anti-rabbit alkaline phosphatase conjugate (Sigma, St. Louis, MO, USA) and BCIP/NBT substrate (Roche, Basel, Switzerland).

### 2.2. Plant Hormone Analysis

Phytohormone content (abscisic acid (ABA); indoleacetic acid (IAA); isopentenyladenine (iP); trans-zeatin (tZ); jasmonic acid (JA); jasmonoyl-isoleucine (JA-Ile); gibberellin A_1_, (GA_1_); gibberellin A_4_, GA_4_; and salicylic acid, SA) was determined by previously reported method [41] with specified modifications. In brief, dry pericarps were homogenized to a fine powder and 50 mg of the sample was suspended in 4 mL of the extraction buffer (1% (*v/v*) acetic acid in acetonitrile/water (4:1)) with a mixture of stable isotope-labeled internal standards (IS) [41]. Suspended samples were extracted for 1 h at 4 °C and centrifuged at 3000× *g* for 10 min at 4 °C. Supernatants were collected, and the pellets were washed with an additional 4 mL of the extraction buffer without IS and centrifuged as before. Acetonitrile from combined supernatants was evaporated in a vacuum concentrator and samples in the 1% aqueous solution were purified by solid phase extraction using Oasis-HLB, -MCX, and -WAX cartridges (Waters Corp., Milford, MA, USA) to obtain acidic (ABA, IAA, JA, JA-Ile, SA, GA1, GA4) and basic (tZ, iP) fractions. In contrast to the previous method, iP and tZ were eluted from the Oasis MCX cartridge by NH_4_OH/water/acetonitrile (1:8:10) after washing with 1.2% (*v/v*) NH_4_OH solution. While SA was previously collected from an aliquot of the Oasis MCX eluate, it was now recovered from the Oasis WAX cartridge by 3% (*v/v*) formic acid in acetonitrile after initial elution of ABA, IAA, JA, JA-Ile, GA_1_ and GA_4_ by 1% (*v/v*) acetic acid in acetonitrile/water (4:1). After evaporation and volume reduction in each fraction, samples were analyzed on the Agilent 1260–6410 Triple Quad LC/MS system (Agilent Technologies Inc., Santa Clara, CA, USA) equipped with a Capcell Pak ADME-HR S2 column (Osaka Soda Co. Ltd., Osaka, Japan). In addition to column type, the gradient of the mobile phases was changed from 3% to 55% in 22 min at a flow rate of 0.4 mL min^−1^ for ABA, IAA, JA, JA-Ile, GA_1_ and GA_4_; for SA, 3% to 98% in 8 min at a flow rate of 0.4 mL min^−1^ was used in the modified method. Gradient conditions for iP and tZ remained unchanged. Mass-to-charge ratio (*m/z*) transitions of analytes were used as described [42]. The contents of the plant hormones were calculated by comparison to respective IS peaks and normalization by dry weight for each sample.

### 2.3. Nutrient Analysis

Fresh powdered pericarp (30 mg) or 30 seeds of *Brassica juncea* from untreated (control) and stress-treated plants (salt, HS, SHS) were incubated with 600 μL Milli-Q for 14 h on an orbital shaker at 4 °C. After incubation samples were centrifuged at high speed (16,000× *g*) and the supernatant was collected, filtered through 0.22 μm spin filter and 200 μL of each sample were diluted with 5.8 mL of Milli-Q water and subjected to nutrient analysis by the inductively coupled plasma-optical emission spectroscopy (ICP-OES) using ICP-720-ES (Varian Inc., Palo Alto, CA, USA). Ca, Mg, Cl, Na, P, S, K were released upon hydration from the mustard pericarp, and also from the seed release as determined by the ICS-5000 instrument (Dionex, Thermo Fisher Scientific Sunnyvale, CA, USA). Data were analyzed by Chromeleon 6.8 chromatography data system (Dionex, Thermo Fisher Scientific, Sunnyvale, CA, USA).

### 2.4. Germination Assays

Germination of *B. juncea* seeds were performed in four replicates, each containing 20 seeds either on Hamra red sandy soil [40], or on a blot paper supplemented with water. Germination was inspected and recorded daily in a course of 4 days. The effect of the extracts obtained from *B. juncea* pericarps on seed germination of *B. juncea* and *Arabidopsis thaliana* and tomato (*Solanum lycopersicum*) was performed in a Petri dish on a blot paper supplemented with water or with control and stress-treated pericarp extracts (10 mg/1 mL water). Germination was initially performed in the dark at 22 °C, and was inspected daily and photographed.

### 2.5. Priming Experiments

One gram of ground pericarps from the control and salt-treated plants of *Brassica juncea* was extracted in 10 mL of water at 4 °C for 12 h with constant rotation. Samples were centrifuged (10 min, 14,000 rpm) and supernatants were collected and used immediately for priming experiments or kept frozen at −20 °C. Seeds of *B. juncea* or grains of a common wheat (*Triticum aestivum*) were imbibed at room temperature for 12 h in 5 mL of water, or 5 mL extracts derived from control and salt pericarps. Seeds/grains were dried out briefly and sown at 1 cm depth on gardening soil in 1 L pots (four seeds of *B*. *juncea* in each pot) or 250 mL pots (two grains in each pot). *B. juncea* was grown in a net house for two weeks followed by two weeks in a controlled room (22 °C +/− 2 °C, 14/10 h day/night photoperiod), while wheat was grown for two weeks in a growth chamber (BINDER Growth chamber KBWF 720, Germany) under 16/8 h day/night photoperiod, 70% humidity and 22/17 °C day/night temperature. Measurements of the dry (60 °C for 24 h) and fresh weights of seedlings and the root system were recorded. Priming assays with *B. juncea* and wheat were repeated two and three times, respectively.

### 2.6. Bacterial Growth Assay

The assay was performed essentially as described [43]. Briefly, *Escherichia coli* (Gram negative) and *Staphylococcus aureus* (Gram positive) were grown overnight on LB medium at 37 °C, then the culture was diluted, transferred to 25% LB and grown at 37 °C to 0.03–0.05 optical density (OD_595_; Epoch, Biotek, Winooski, VT, USA). To a 150 μL aliquot of the culture, we added 50 μL of PBS (control) or 50 μL pericarp extracts (filtered through a 0.22 μM spin filter) three replicates per treatment in a flat-bottom 96-well microtiter plate. Kanamycin (final concentration of 50 μg/mL) was used as a negative control. Plates were incubated in the dark using a spectrophotometer (Synergy 4, Biotek, Winooski, VT, USA) and reads (OD_595_) were taken in intervals of 30 min in a course of 12 h. The average OD for each blank replicate at a given time point was subtracted from the OD of each replicate treatment at the corresponding time point and standard errors were calculated for each treatment at every time point.

### 2.7. Statistical Analysis

Most data collected from the experiment were subjected to an analysis of variance (two-way ANOVA) using the VassarStats software. The difference between means were computed by Post Test Calculator (Graph Pad) at *p* < 0.05. For the comparison of multiple groups, we also used the one-way ANOVA calculator, with Tukey HSD (https://www.socscistatistics.com/tests/anova/default2.aspx; accessed on 13 November 2018).

## 3. Results

### 3.1. Exposure to Salinity and Heat Stress Has a Dramatic Impact on Progeny Seed Production of Brassica juncea

The crop plant *B. juncea* was grown in a net house and subjected gradually, at the time flowering commenced, to increasing concentrations of salt (final concentration 200 mM). Thereafter, half of the control and salt-treated plants were exposed to three heat shock (HS) intervals, each for 3 h at 37 °C in a course of 4 days, after which all plants were irrigated with water until fruits matured and dried out. Figure 1A shows fruits and seeds obtained from the control and stress-treated plants. We verified the response of the plants to HS treatments by immunoblotting with an antibody to the small heat shock protein 17.6 (sHSP17.6) which showed strong upregulation in HS-treated leaves of small HSPs; HSP70 was abundant in all treatments (Figure 1B). The average weights of the fruits of all stress-treated plants were significantly reduced (Figure 1C); yet, pretreatment with salt seemed to mitigate the effect of HS on fruit weight. While the average weight of a seed was reduced (~2-fold) under salt treatment (Figure 1D), the most prominent effect was the complete abortion of seeds derived from plants subjected during flowering and seed filling to HS and SHS (Figure 1E). Seeds produced on salt-treated plants were poorly germinated either on a blot paper (~7.5%) or in gardening soil (5%) compared with seeds from the control plants (93.75% and 82.5% on blot paper and soil, respectively) (Figure 1F).

### 3.2. The Effect of Pericarp Extracts on Seed Germination

Focusing on pericarps, we first sought to examine the effect of the maternal growth conditions on pericarp properties and the capability to control seed germination of *B. juncea* and heterologous species. To this end, seeds of *B. juncea* were germinated on red sandy soil in the presence of pericarp extracts derived from control, salt (S), HS and SHS treated plants and germination was inspected daily up to 4 days after sowing. The results showed that the most notable effect on germination was exerted by pericarps derived from salt and SHS-treated plants showing a significant reduction in the germination rate as well as in final germination after 96 h compared with germination in water or in control pericarp extracts (Figure 2A).

Further germination assays revealed that pericarps from control and stress-treated plants possess allelopathic substances that strongly inhibited the seed germination of *Arabidopsis thaliana* and tomato (*Solanum lycopersicum*) (Figure 2B,C). However, germination was almost fully recovered after washing out the pericarp extracts (Figure 2B,C).

### 3.3. Single and Combined Stresses Altered Phytohormone Accumulation in Dead Pericarps

Pericarps derived from control and stress-treated plants were subjected to a phytohormones analysis by LC-MS. Results showed (Figure 3) that multiple phytohormones including indole acetic acid (IAA), abscisic acid (ABA), gibberellic acid (GA), jasmonic acid (JA) and salicylic acid (SA) as well as the cytokinins, isopentenyladenine (iP) and trans-Zeatin (tZ) were accumulated in the dead pericarps and their levels were significantly altered under stress conditions. Thus, the levels of IAA and ABA were reduced significantly under salt stress and further reduced under combined SHS stresses. GA1 which was absent in the control, and HS pericarps were up-accumulated under salt stress and further enhanced under combined SHS stresses. The accumulation of other phytohormones, GA4, JA and SA was increased significantly under salt stress, but abolished under combined SHS stresses. The cytokinin tZ was up-accumulated in pericarps of all stress-treated plants. Notably, SA was accumulated to high levels under HS or salt treatment but reduced to the control levels under combined SHS stresses.

### 3.4. Single and Combined Stresses Altered Nutrient Levels Extracted from Pericarps or Secreted from Seeds

We used the inductively coupled plasma (ICP) for analysis of the nutrients released from the seeds or extracted from pericarps derived from the control or stress-treated plants. Results showed (Figure 4) that the pericarps of *B. juncea* store and release upon hydration multiple nutritional elements. We observed an increase in the accumulation of Ca, P and S in pericarps of stressed plants, an increase in Mg under salt and SHS, while K was increased only under salt treatment. Interestingly, progeny seeds derived from salt-treated plants accumulated high levels of nutritional elements including Mg, P, S and K, which were significantly reduced in progeny seeds derived from SHS-treated plants. As expected, pericarps derived from salt-treated plants accumulated high levels of sodium (Na) and chlorine (Cl).

### 3.5. Pericarps Possess Bacterial Growth Promoting Substances

We examined the potential of pericarp extracts from treated plants to control microbial growth. To this end, *Escherichia coli* and *Staphylococcus aureus* were grown in a flat-bottom 96-well microtiter plate in LB medium supplemented with PBS or with pericarp extracts of control and stress-treated plants. Ampicillin and kanamycin were used as antibiotic references for *E. coli* and *S. aureus,* respectively. Plates were incubated in the dark using a Synergy 4 spectrophotometer and reads (OD_595_) measurements were taken at 30 min intervals over a course of 12 h. Results showed (Figure 5) that the growth of both *S. aureus* (Figure 5A) and *E. coli* (Figure 5B) was accelerated significantly in the presence of pericarp extracts irrespective of their source. As expected, both ampicillin and kanamycin completely inhibited the growth of the bacteria.

### 3.6. Priming Capacity of Dead Pericarps

Phytohormones such as IAA, ABA and SA are commonly used to prime seeds for improving germination, seedling growth and development and tolerance to abiotic stresses [33]. The finding that phytohormones are accumulated in dead pericarps, many are known to have priming capacity, prompted us to investigate the effect of the maternal environment on the priming capacity of dead pericarps. To this end, *B. juncea* seeds were imbibed for 12 h with extracts of pericarps derived from treated plants followed by sowing in standard gardening soil; seedling performance was recorded after 4 weeks. While seeds germinated similarly under all treatments, seedling performance was differently affected by the treatments. Accordingly, seedlings derived from seeds imbibed in pericarp extracts performed better (Figure 6). Most notable was the high development of the root system of seedlings derived from seeds pretreated with the control pericarp extract (Figure 6A), which is reflected by the average root dry weight per plant (Figure 6B).

In another experiment, seeds of a heterologous species, namely grains of cultivated wheat (*Triticum aestivum*) were imbibed with pericarp extracts for 12 h after which they were sown in soil and their emergence and growth parameters were examined. We observed initial radicle protrusion in wheat grains imbibed in water and control pericarp (Cont-P) extract but not in grains imbibed in Salt-P extract (Figure 7A). However, all seedlings emerged simultaneously 3 days after sowing. After 2 weeks seedlings were examined for their growth and development. Figure 7B shows that pre-treatment with Cont-P had a negative effect on seedling performance having reduced root length compared with water and Salt-P. All growth parameters including seedling and root fresh and dry weights were negatively affected by Cont-P (Figure 7 C–F), while Salt-P appeared to significantly increase root dry weight compared with water and Cont-P (Figure 7F).

## 4. Discussion

We will describe the considerable impact of salinity and short-term exposure to heat stress during the reproductive phase of the crop plant *B. juncea* on progeny seed and particularly on dead pericarp properties. Our data showed that changes in the maternal environment and particularly short episodes of high temperature at the reproductive phase have a dramatic impact on embryo development and the properties of dead pericarps. The study also highlighted that temperature fluctuations are most important in determining crop production since short exposure (3 h) of plants to a high temperature (37 °C) during the reproductive phase was devastating, resulting in complete seed abortion. The adverse effect of high temperatures at the reproductive stage on *Brassica* crop yield is well-documented. Accordingly, exposure of *Brassica napus* plants to short-term heat shock treatments (4 h at 35 °C daily, in a course of 1 or 2 weeks) during the reproductive stage resulted in a substantial loss of yield [44] and exposure of the *Brassica* species to a high temperature (35/15 °C light/dark) for 7 days resulted in 89% reduction in mean seed yield on the main stem [45]. *B. napus* plants grown continuously at 27/17 °C light/dark were almost entirely sterile [46]. The assessment of the effect of global warming on crop yield predicted that an increase in global mean temperature by one degree Celsius might reduce significantly the yields of major crops including wheat, rice, maize and soybean [47]. Yet, the most detrimental effect on yield is the exposure of crops during the reproductive stage to short episodes of high temperatures (heat wave/warm spell) [48,49,50,51]. Thus, the heat waves and warm spells that are predicted to increase in frequency in many regions of the world [7,52,53] pose a serious threat on global food production.

The effect of the maternal environment on seed and dead pericarp properties of *B. juncea* are consistent with recent published data demonstrating the impact of abiotic stresses on the dispersal unit properties, particularly the properties of the DOEEs of wild plants such as *Anastatica hierochuntica* and *Avena sterilis* [22,23].

Plant response to a combination of stresses has been intensively studied during the last two decades confirming the idea that a combination of stresses is interpreted by plants as a peculiar stress condition that elicits a distinctive response [9,14,15,16,17]. This is well-demonstrated by the effect of single and combined heat and salt stress on the accumulation of phytohormones as well as nutritional elements in dead pericarps. Thus, dead pericarps of *B. juncea* accumulate various phytohormones whose levels were significantly affected by exposure of the mother plants, at the reproductive stage, to single and combined stresses. Accordingly, we observed significant reduction in IAA and ABA content in pericarps derived from plants exposed to salt, which was intensified in combination with HS. Similarly, the accumulation of phytohormones in pericarps of the desert plant *Anastatica hierochuntica* was affected by salt [22]. On the other hand, the phytohormone GA1, which is absent in the control and HS pericarps was significantly elevated under salt treatment and further enhanced in combined SHS pericarps. However, while salt treatment enhanced the accumulation of JA, JA-Ile and SA in pericarps, these phytohormones were down-accumulated in pericarps derived from combined SHS-treated plants. Commonly, seed germination is known to be regulated antagonistically by ABA and GA, which inhibit and promote germination, respectively [54]. Yet, a decrease in ABA and an increase in GA1 levels in salt and SHS pericarps did not relieve the pericarp extract’s inhibitory effect on the germination of heterologous species. We assume that although ABA was decreased in S and SHS pericarps, its level (300–500 ng/gDW) was sufficient to exert an inhibitory effect on germination, while the increase in GA level (7–14 ng/gDW) was not sufficient to negate the ABA effect. Alternatively, other substances accumulated in pericarps might be involved in the specific inhibitory effect on seed germination [55]. Notably, SA was accumulated to the highest levels compared with other phytohormones, particularly under single HS or salt stress. SA is a well-known phytohormone that performs key roles in plant immunity [56] and together with other phytohormones including ABA, IAA and ethylene are commonly used in seed priming to enhance seed performance and fate particularly under stress conditions [33]. Thus, single and combined stresses differently affected the accumulation of phytohormones in dead pericarps of *B. juncea*. These phytohormones might be released upon hydration to the immediate surroundings of the seeds and prime them to ensure their success in the ecological niche. Generally, seed priming is an agricultural practice whereby seeds are hydrated with multiple substances to achieve the cellular state of germination without radicle protrusion, which often leads to improved plant performance [33]. Seed priming with phytohormones is an often-used technique to improve seed germination and seedling establishment as well as to confer stress tolerance and eventually enhance crop yield [33,57]. Interestingly, the pre-treatment of seeds with dead pericarp extracts affected seedling performance in a species-specific manner. Thus, control pericarp extracts possess priming activity on *B. juncea* seedlings, which is reduced following the exposure of plants to salt stress. However, an inverse effect was observed on a heterologous species (wheat) whereby pretreatment with control and salt pericarp extracts had a negative and positive effect on seedling performance, respectively. Thus, under normal growth conditions, substances are accumulated in dead pericarps to increase the survival rate of germinating seeds by priming and improving seedling performance on the one hand, and attenuating the growth of heterologous species on the other hand. However, under stress conditions, pericarps negatively affected autologous species, but were harmless or beneficial to heterologous species. Taken together, the strategy employed by the plant at the early stages of development for securing resources (e.g., allelopathy, cooperativity) is set up, at least partly, by the mother plants in conjunction with the maternal environment whose components are stored in the dead maternal organs enclosing the embryo.

## 5. Conclusions

The data presented here highlighted the enormous impact that single and combined stresses during the reproductive phase might have on the functional properties of dead pericarps as well as on seed quantity and quality. Dead pericarps of *B. juncea,* commonly considered as agricultural waste, appear to function as a rich storage for multiple beneficial substances such as growth factors and nutritional elements and whose levels, composition and priming activity are changed following the mother plants exposure to single and combined stresses. These data further highlighted the importance of the dead organs enclosing the embryo in seed biology and ecology and consequently in plant population dynamics and diversity. The detrimental effect of stress on yield highlights the reproductive stage as the most vulnerable in the face of climate change, which might have implications for global food security.

## Figures and Tables

**Figure 1 plants-10-01627-f001:**
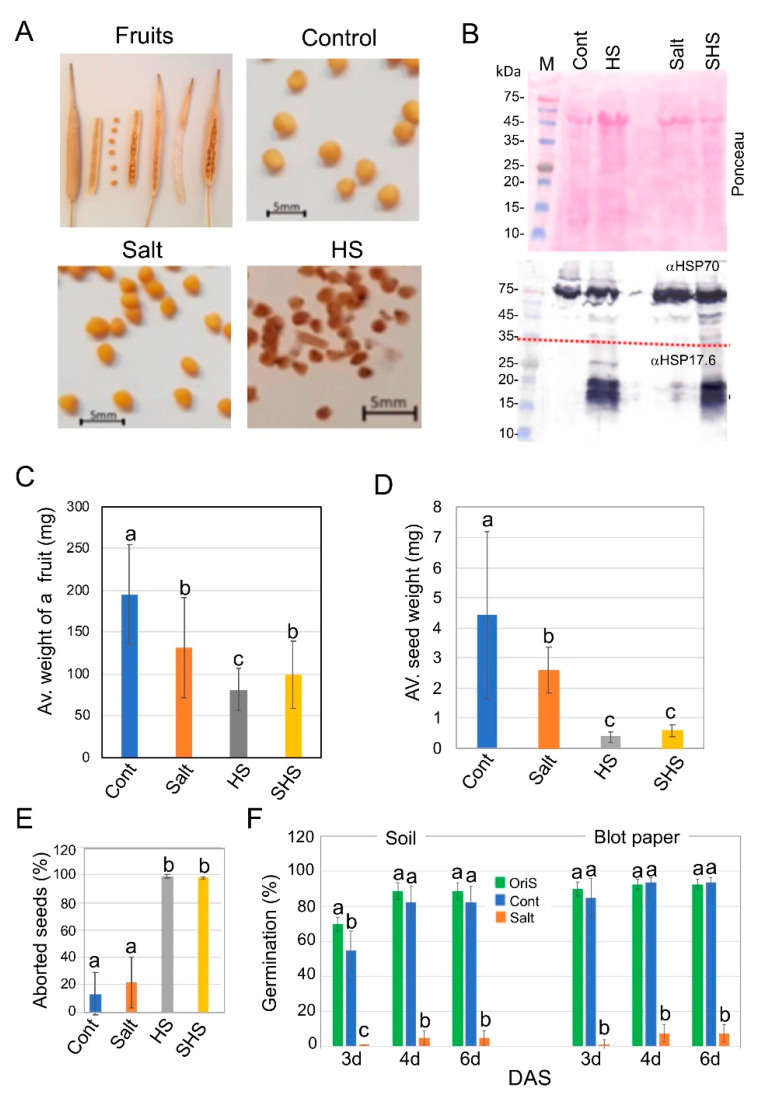
Effect of maternal environment on seed performance and fate of *B. juncea*. (**A**) Mature fruits and seeds of *B. juncea*. Note the aborted seeds in HS-treated plants. (**B**) Heat shock induces expression of small HSPs. Proteins extracted from leaves of control (Cont) or heat shock (HS), salt and SHS-treated plants were subjected to immunoblotting (lower panel) using anti-HSP70 (αHSP70) and anti-HSP17.6 (αHSP17.6). Upper panel is the Ponceau staining of the membrane. Note the membrane was cut into two parts (broken line), the upper containing proteins above 35 kDa was probed with αHSP70 and the lower part with αHSP17.6. M, protein molecular weight markers given in kDa. (**C**) Average weight of a fruit. (**D**) Average weight of a seed. (**E**) Percentage of aborted seeds. Note the complete abortion of seeds under heat shock (HS) treatments. (**F**) Germination of seeds derived from slat-treated plants is significantly reduced compared with control (Cont) and original seed stock (OriS). Germination was performed on red sandy soil (Soil) or on a blot paper. DAS: days after sowing. Vertical bars represent the standard deviation. Different letters indicate statistically significant differences between treatments (*p* < 0.05). Statistical analysis was performed by two-way ANOVA except for panel E which was analyzed by one-way ANOVA calculator plus Tukey HSD.

**Figure 2 plants-10-01627-f002:**
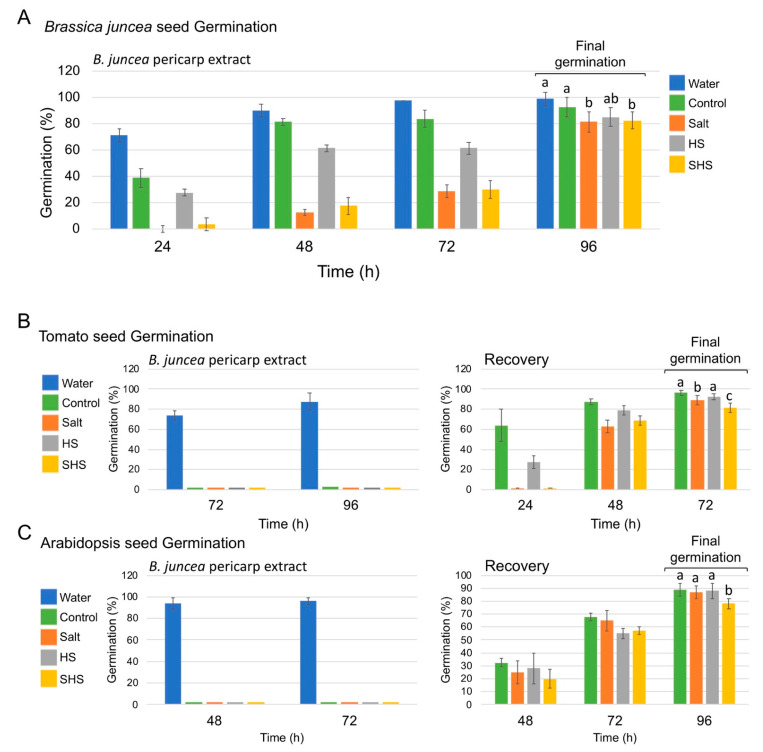
Effect of pericarp extracts on seed germination of *B. juncea* and heterologous species. Pericarps derived from control plants or plants exposed to salt, HS and SHS were extracted and analyzed for their effect on seed germination of *B. juncea* (**A**), tomato (*Solanum lycopersicum*) (**B**) and *Arabidopsis thaliana* (**C**) in comparison to water. Germination was inspected at different times after sowing and recorded as percentage of germination. Cont: control plants irrigated with water; HS: heat shock; S: salt. Vertical bars represent the standard deviation. Statistical significance was performed for final germination using two-way ANOVA Calculator (VassarStats). Different letters indicate statistically significant differences between treatments (*p* < 0.05).

**Figure 3 plants-10-01627-f003:**
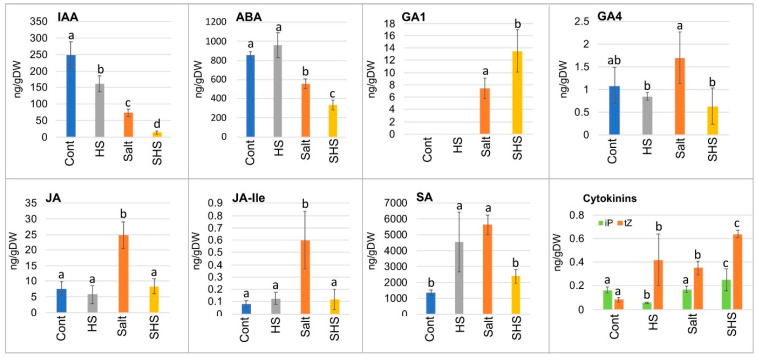
Dead pericarps accumulate multiple phytohormones whose levels are modified under stress conditions. Content of phytohormones in pericarps derived from control and the indicated stress conditions are given in nanogram per gram dry weight (ng/gDW). IAA: indole acetic acid; ABA: Abscisic acid; GA1 and GA4: Gibberellic acid 1 and 4; JA: jasmonic acid; JA-Ile: JA-isoleucine; SA: salicylic acid; iP: isopentenyladenine; tZ: trans-Zeatin. Vertical bars represent the standard deviation. Different letters indicate statistically significant differences between treatments (*p* < 0.05). Statistical analysis was performed by one-way ANOVA Calculator plus Tukey HSD (Social Science Statistics).

**Figure 4 plants-10-01627-f004:**
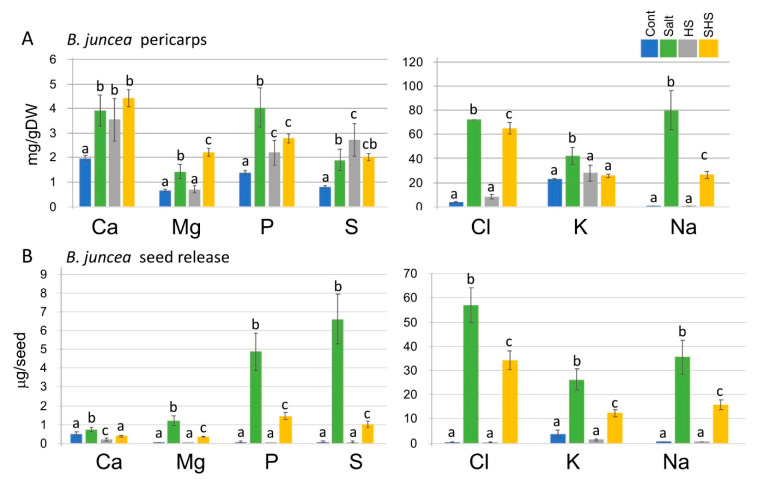
Effect of maternal environment on nutrient levels. Nutrient levels in *B. juncea* pericarp extract (**A**) or released from seeds (**B**) are altered following exposure of mother plants to stress. Pericarps and seeds derived from control (Cont), salt (S), heat shock (HS) and SHS-treated plants were subjected to nutrient detection by ICP-OES. The concentration of each element was calculated either as milligram (mg) per gram dry weight (gDW) or as microgram (μg) per seed. Vertical bars represent the standard deviation. Different letters indicate, for each element, statistically significant differences between treatments (*p* < 0.05). Note that salt-treated plants display (as expected) high levels of chlorine (Cl) and sodium (Na) in pericarps and seeds.

**Figure 5 plants-10-01627-f005:**
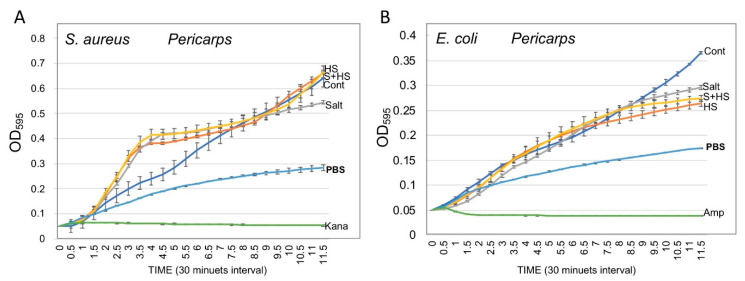
Pericarps of *B. juncea* contain substances that promote bacterial growth. *Staphylococcus aureus* (**A**) and *Escherichia coli* (**B**) were grown in a flat-bottom 96-well microtiter plate in the presence of PBS (control), or in the presence of substances extracted from pericarps derived from control and stress-treated plants. Kanamycin (Kana) and ampicillin (Amp) were used as antibiotic references. Bacterial growth was monitored by measuring the OD_595_ of the culture at 30 min intervals over the course of 12 h. Each treatment was performed in triplicates and vertical bars represent the standard deviation.

**Figure 6 plants-10-01627-f006:**
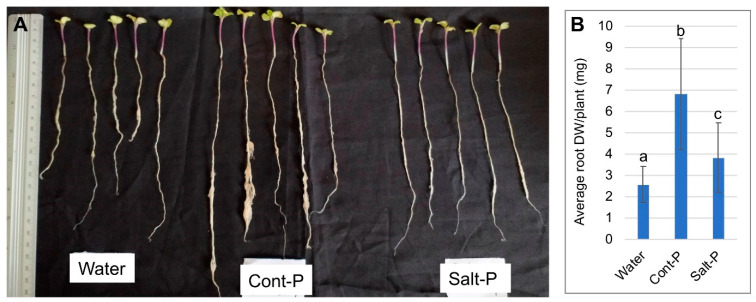
Priming *B. juncea* seeds with pericarp extracts improved their performance. *B. juncea* seeds were imbibed with water or pericarp extracts from control or salt-treated plants followed by drying the seeds, sowing and inspection of seedling growth. (**A**) Seedling samples derived from seeds pretreated with water or pericarp extracts from control (Cont-P) and salt-treated plants (Salt-P). (**B**) Average root dry weight (DW) per plant. Vertical bars represent the standard deviation. Different letters indicate statistically significant differences between treatments (*p* < 0.05) (*n* = 50). Statistical analysis was performed by one-way ANOVA Calculator plus Tukey HSD (Social Science Statistic).

**Figure 7 plants-10-01627-f007:**
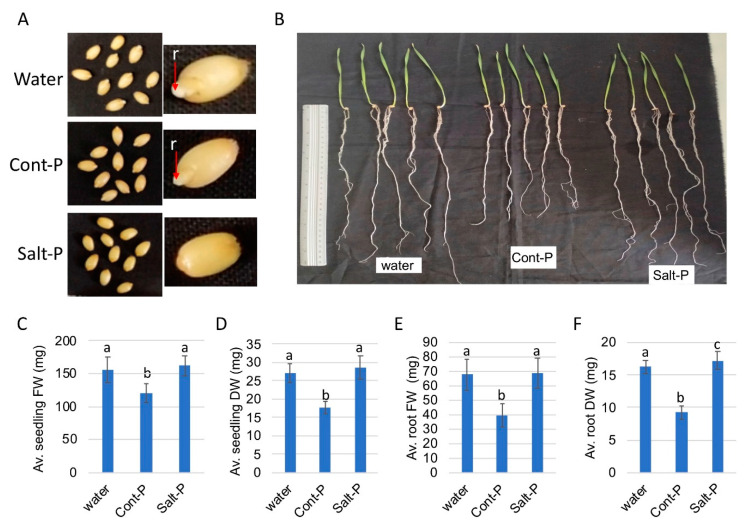
Effect of *B. juncea* pericarp extracts on wheat seedling performance. Wheat grains were imbibed with water or pericarp extracts from control (Cont-P) or salt (Salt-P)-treated plants followed by sowing and inspection of seedling growth. (**A**) Wheat grains after imbibition. Note the initial radicle protrusion (r, red arrow) in water and Cont-P. (**B**) Seedling samples derived from wheat grains pretreated with water, Cont-P and Salt-P. (**C**) Average seedling fresh weight (FW). (**D**) Average seedling dry weight (DW). (**E**) Average root FW per seedling. (**F**) Average root DW per seedling. Vertical bars represent the standard deviation. Different letters indicate statistically significant differences between treatments (*p* < 0.05) (n = 26). Statistical analysis was performed by one-way ANOVA Calculator plus Tukey HSD (Social Science Statistic).

## Data Availability

The data that support the findings of this study are available in the main text.

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
