# Peer review of "Single and Combined Salinity and Heat Stresses Impact Yield and Dead Pericarp Priming Activity"

_plants, 2021, doi:10.3390/plants10081627_

Round 1

Reviewer 1 Report

With respect to the manuscript that I reviewed entitled: Single and combined salinity and heat stresses impact yield and dead pericarp priming activity, you will be interested in the following appreciations:

Abstract: concise, descriptive, emphasizing the idea, aim of the present study.  

Introduction: is well written, well documented with relevant literature (very actual, from the last 10 years) and it is able to introduce and familiarize with the subject of the study, in a very gradual and logical way.

Materials and Methods: clear, logical and easily replicable. Perhaps is a little bit large, but I believe that this is the authors ‘choice to describe in very details the used methods, comparatively to others, who only refer to the standard method, without giving many technical details. However, this is not in any case a weak point, but only a simple observation.

Regarding the section results is presented in a concise, easy to follow manner, also using the 7 figures (14 graphs and 8 photos), which are very well organized.

The discussion section is concise, compared to the research of other authors. The statements are clearly supported by data and are linked to the paper's goal. Study implications and limitations are completely and succinctly presented.

Conclusions: concise and very well presented

References: correlated well with the text

Author Response

Thanks, we have no further response.

Reviewer 2 Report

In this study, authors investigated the effects of salt, heat and their combination on the characteristics of dead pericarp of B. juncea. Authors demonstrated the alteration of various hormones and nutrients in dead pericarps subjected to stresses. These dead pericarps inhibit seed germination of B. juncea, but, accelerate bacterial growth and seedling growth of B. juncea.

This study showed potential of application of dead pericarp to control crop growth. However, authors should address some comments below prior to the publication.

Authors performed immunoblot analysis and present the data in figure 1. But, the methods are not explained in the “Materials and Methods”. I am not sure if antibodies of proteins in B. juncea are commercially available or not. Thus, authors should explain more detailed information of antibody.

In figure 1F, germination of seeds derived from the plants exposed to heat and S+H plant is not presented. I think authors did not perform the experiment because seed development was too much abolished under these conditions (Figure 1E). Authors should explain it in the text or figure legend.

Authors discussed effects of hormones on germination in the “Discussion” section. However, I feel that many readers still wonder why salt and S+H treated pericarps inhibit germination, because ABA decreased, but, GA increased. ABA and GA are very important hormones for dormancy and germination, respectively.

Author Response

Reviewer 2

In this study, authors investigated the effects of salt, heat and their combination on the characteristics of dead pericarp of B. juncea. Authors demonstrated the alteration of various hormones and nutrients in dead pericarps subjected to stresses. These dead pericarps inhibit seed germination of B. juncea, but, accelerate bacterial growth and seedling growth of B. juncea.

This study showed potential of application of dead pericarp to control crop growth. However, authors should address some comments below prior to the publication.

Authors performed immunoblot analysis and present the data in figure 1. But, the methods are not explained in the “Materials and Methods”. I am not sure if antibodies of proteins in B. juncea are commercially available or not. Thus, authors should explain more detailed information of antibody.

Detailed experimental procedure of immunoblotting is given in M&M. [page 6]

In figure 1F, germination of seeds derived from the plants exposed to heat and S+H plant is not presented. I think authors did not perform the experiment because seed development was too much abolished under these conditions (Figure 1E). Authors should explain it in the text or figure legend.

Indeed, under heat stress, seeds were completely aborted. This is indicated in the text. [page 10]

Authors discussed effects of hormones on germination in the “Discussion” section. However, I feel that many readers still wonder why salt and S+H treated pericarps inhibit germination, because ABA decreased, but, GA increased. ABA and GA are very important hormones for dormancy and germination, respectively.

We discussed further ABA and GA and their relationship to germination. [page 21]

Reviewer 3 Report

In their article "Single and combined salinity and heat stresses impact yield and dead pericarp priming activity", Swetha et al. present a well-conducted study on the impact of salt, heat, and a combination thereof on the yield of Brassica juncea. The authors have carried out a lot of work and evaluated a series of relevant parameters such as the impact on seed and fruits weight and health.
Overall, this is a solid piece of work.

I just have a few remarks regarding the text and, importantly, data availability.

The authors have analysed a series of major phytohormones in their study. However, in the introduction there is no explanation of what these phytohormones are / why phytohormones are important. This might sound trivial, but in general this work lacks a bit of connection to a molecular plant science level. Some reviews on how phytohormones orchestrate and balance the response to the environment with growth should be cited. Concepts such as the plant perceptron (see "The plant perceptron connects environment to development" in Nature volume 543, pages 337–345 (2017) doi: 10.1038/nature22010 )should be introduced.

"Crop plants are essentially highly sensitive to abiotic stresses that cause significant yield losses worldwide [3, 28,29]. Multiple studies related to the effect of heat shock on plant performance were performed under long-term exposure (often >24 h) to high tem-peratures (37-45oC), though in recent years the effect of short episodes of high temperature (heat waves/hot spells) are getting more attention [17,30]."
-> This almost reads as if the responses to heat stress were a crop-specific thing. I think it is important to highlight that (a) a lot of what we know about plant model systems like Arabidopsis (which is not a crop) and (b) that these responses appear evolutionarily conserved. Indeed, some responses can be found across the land plant tree of life and even in the losest algal relatives of land plants. 
Please read and cite "Heat stress response in the closest algal relatives of land plants reveals conserved stress signaling circuits" doi: 10.1111/tpj.14782 -- Plant J 2020 Aug;103(3):1025-1048
As well as
"On the evolution of plant thermomorphogenesis" doi: 10.1093/jxb/erab310 -- Journal of Experimental Botany, erab310

In order to adhere to open data practices, please upload / deposit the mass spectrometry data onto metabolights, https://www.ebi.ac.uk/metabolights/

Author Response

Reviewer 3

In their article "Single and combined salinity and heat stresses impact yield and dead pericarp priming activity", Swetha et al. present a well-conducted study on the impact of salt, heat, and a combination thereof on the yield of Brassica juncea. The authors have carried out a lot of work and evaluated a series of relevant parameters such as the impact on seed and fruits weight and health.

Overall, this is a solid piece of work.

I just have a few remarks regarding the text and, importantly, data availability.

The authors have analysed a series of major phytohormones in their study. However, in the introduction there is no explanation of what these phytohormones are / why phytohormones are important. This might sound trivial, but in general this work lacks a bit of connection to a molecular plant science level. Some reviews on how phytohormones orchestrate and balance the response to the environment with growth should be cited. Concepts such as the plant perceptron (see "The plant perceptron connects environment to development" in Nature volume 543, pages 337–345 (2017) doi: 10.1038/nature22010 )should be introduced.

We added to the Introduction section a paragraph regarding phytohormones and seed priming. [page 4]

"Crop plants are essentially highly sensitive to abiotic stresses that cause significant yield losses worldwide [3, 28,29]. Multiple studies related to the effect of heat shock on plant performance were performed under long-term exposure (often >24 h) to high tem-peratures (37-45oC), though in recent years the effect of short episodes of high temperature (heat waves/hot spells) are getting more attention [17,30]."

This almost reads as if the responses to heat stress were a crop-specific thing. I think it is important to highlight that (a) a lot of what we know about plant model systems like Arabidopsis (which is not a crop) and (b) that these responses appear evolutionarily conserved. Indeed, some responses can be found across the land plant tree of life and even in the losest algal relatives of land plants.

Please read and cite "Heat stress response in the closest algal relatives of land plants reveals conserved stress signaling circuits" doi: 10.1111/tpj.14782 -- Plant J 2020 Aug;103(3):1025-1048

As well as "On the evolution of plant thermomorphogenesis" doi: 10.1093/jxb/erab310 -- Journal of Experimental Botany, erab310

We rephrased the sentence to indicate the general response of land plants to abiotic stresses. [page 4]

The reference de Vries et al., 2020 was added [reference # 37.]

In order to adhere to open data practices, please upload / deposit the mass spectrometry data onto metabolights, https://www.ebi.ac.uk/metabolights/

The metabolomic data is available in supplementary material.